# Gut Metabolites and Breast Cancer: The Continuum of Dysbiosis, Breast Cancer Risk, and Potential Breast Cancer Therapy

**DOI:** 10.3390/ijms23169490

**Published:** 2022-08-22

**Authors:** Kayla Jaye, Dennis Chang, Chun Guang Li, Deep Jyoti Bhuyan

**Affiliations:** NICM Health Research Institute, Western Sydney University, Penrith, NSW 2751, Australia

**Keywords:** gut microbial metabolites, breast cancer, nisin, inosine, butyrate, cancer, sodium butyrate, standard chemotherapy

## Abstract

The complex association between the gut microbiome and cancer development has been an emerging field of study in recent years. The gut microbiome plays a crucial role in the overall maintenance of human health and interacts closely with the host immune system to prevent and fight infection. This review was designed to draw a comprehensive assessment and summary of recent research assessing the anticancer activity of the metabolites (produced by the gut microbiota) specifically against breast cancer. In this review, a total of 2701 articles were screened from different scientific databases (PubMed, Scopus, Embase and Web of Science) with 72 relevant articles included based on the predetermined inclusion and exclusion criteria. Metabolites produced by the gut microbial communities have been researched for their health benefits and potential anticancer activity. For instance, the short-chain fatty acid, butyrate, has been evaluated against multiple cancer types, including breast cancer, and has demonstrated anticancer potential via various molecular pathways. Similarly, nisin, a bacteriocin, has presented with a range of anticancer properties primarily against gastrointestinal cancers, with nominal evidence supporting its use against breast cancer. Comparatively, a natural purine nucleoside, inosine, though it has not been thoroughly investigated as a natural anticancer agent, has shown promise in recent studies. Additionally, recent studies demonstrated that gut microbial metabolites influence the efficacy of standard chemotherapeutics and potentially be implemented as a combination therapy. Despite the promising evidence supporting the anticancer action of gut metabolites on different cancer types, the molecular mechanisms of action of this activity are not well established, especially against breast cancer and warrant further investigation. As such, future research must prioritise determining the dose-response relationship, molecular mechanisms, and conducting animal and clinical studies to validate in vitro findings. This review also highlights the potential future directions of this field.

## 1. Introduction

Gut microbiota is the collective term for microorganisms residing within the human gut and they interact with the host immune system both directly and indirectly to contribute to gut homeostasis and fight infection [1]. The metabolic by-products of gut microbiota, gut metabolites have demonstrated a close interrelationship with the host in the inhibition of different cancer types, including breast cancer [2]. In recent years, studies have explored the diverse biological and anticancer properties of the metabolites produced by the gut microbiota against different cancer types. In contrast, dysbiosis of the gut microbial population has been shown to contribute to the onset of different diseases including breast cancer and colorectal cancer (CRC) [3] (Figure 1). Therefore, further investigation into the broad spectrum of activities of gut microbial metabolites is crucial for their use as potential natural anticancer agents.

### 1.1. The Gut Microbiota and Its Metabolites

The gut microbiota consists of 100 trillion microorganisms residing in the gut, which is inclusive of bacteria, protozoa, fungi, and viruses [1]. Each individual houses a unique gut microbiome composition, which is determined by a range of lifestyle and genetic factors, including diet, physical activity, antibiotic use, and the maternal microbial composition (Figure 2) [1,4]. The development of a healthy gut microbiota population commences in the perinatal stage of an infant and continues into the postnatal phase [4]. This composition is influenced by factors such as gestation duration, maternal nutrition, and antibiotic consumption in the perinatal phase, and other factors including feeding type, i.e., breastfed or formula-fed, exposure to family members, and antibiotic use in the postnatal phase (Figure 2) [4]. The maternal microbiome holds strong significance in encouraging the development of a strong immune system in a newborn, as well as influencing the infant’s health in various life stages [5]. In the evaluation of human breast milk, it was observed that there was a high concentration of butyrate-producing bacterial species, including *Faecalibacterium* and *Coprococcus*, which may indicate a correlation between improved infant health and the presence of short-chain fatty acids (SCFAs) [5]. Further evolution and maturation of the gut microbial composition in infants and young children are influenced by exposure to pathogenic microbes and the introduction of new microorganisms with a changing diet [4]. There is substantial evidence supporting the further contribution of an individual’s diet to their gut microbial composition throughout their lifetime. It has been well-established that a diet rich in fibre and resistant starch can have protective effects against cancer, as both are fermented by the gut microbiota, leading to the production of metabolites, SCFAs, as by-products [1]. The three most produced SCFAs of this metabolic process include butyrate, propionate, and acetate, each of which has demonstrated anti-inflammatory properties within the host in preclinical studies [6]. Interestingly, the Westernized diet has been observed to disrupt the healthy gut microbial population, due to the high intake of red and processed meats and low intake of dietary fibre and key vitamins [6]. In particular, a Westernized diet can result in the metabolism of toxic gut metabolites that can replace healthy bacterial species in the gut microbiome [6]. The modification of dietary patterns has demonstrated a causative correlation with the onset or prevention of several diseases, which is a significant factor of consideration in the maintenance of gut homeostasis and the treatment of gut dysbiosis.

The complex connection between gut microbiota and host health can be further understood by the gut-brain axis (GBA) and its associated biological activities. The GBA is constituted by the gut, central nervous system (CNS), enteric nervous system (ENS), autonomic nervous system, entero-endocrine system, and the hypothalamic-pituitary-adrenal axis, which collectively assist in the secretion of hormones and neuro-hormones to support in the maintenance of complex metabolic processes [1]. The administration of probiotics has been shown to aid in the maintenance of gut homeostasis which further impacts other major body systems within the GBA, as indicated by an increase in immune-modulatory action following the consumption of probiotics [7]. The cumulative effects of gut microbial metabolites impact various body systems within the GBA, however, the relationships between these metabolites and body systems are correlative, not causative [5]. Gut microbiota also interacts with the CNS and the ENS to modulate nutrient metabolism via the production of neurotransmitters [8]. The gut microorganisms regulate the vagal afferent pathway and intestinal metabolism via the GBA by modulating the levels of specific peptides, including ghrelin, leptin, cholecystokinin, and 5-hydroxytryptamine (serotonin) [8]. The serotonin peptide plays a vital role in the GBA complex, in which disturbances to serotonin concentration levels can interrupt the normal function of signalling pathways along the GBA, which can impact neural processes associated with the gastrointestinal tract (GIT) [8]. In the case of triple-negative breast cancer (TNBC), dysbiotic disturbances to serotonin concentrations have been shown to play a key role in tumour progression [8]. Furthermore, the breast and gut microbiome crosstalk directly via the breast-gut microbiome axis, and the production of specific gut metabolites activate the epithelial-mesenchymal transition (EMT) program and the metabolism of estrogen molecules [9]. Gut microbial metabolites have been shown to impact estrogen receptor-positive tumour sub-types due to this direct regulation of estrogen metabolic processes [9]. As such, the GBA is an important factor of consideration in the action of gut microbial metabolites within the host.

### 1.2. The Crosstalk between Gut Metabolites and the Host Immune System

The close interrelationship between gut microbes, their metabolites, and the host immune system has been acknowledged in the literature. The association between gut microbes and the development of the immune system commences in infancy, in which the transmittance of microflora species during breastfeeding and the metabolic breakdown of complex polysaccharides establishes the infant gut microbial composition and primes the immune system [10]. The primary bacterial components identified as rich in breast milk include Firmicutes species (such as *Clostridium* and *Lactobacillus*), Actinobacteria species (including *Propionibacterium*), and Bacteroidetes species (such as *Prevotella*), which assist in the metabolic production of hormones and gut metabolites [10]. The reduction in microbial diversity and low concentrations of *Bifidobacterium* and *Bacteroides* species can result in an altered stimulation of the immune system and a predisposition to Th2-mediated allergies in infancy [4]. For instance, the low concentrations of *Bifidobacterium* and *Bacteroides* bacterium were observed in C-section infants, leading to Th2-mediated allergies due to a reduction in circulating levels of Th1 chemokines [4]. Probiotic administration and consumption have been shown to modify this altered development and improve the quality of life for patients with allergic rhinitis and high-risk asthma [7]. This was further investigated by an in vivo study that observed that in the absence of protective commensal bacteria, germ-free mice experienced modified development and immunologic defects of both the adaptive and innate immune systema [11]. As a primary level interaction, the gut microbial species can exhibit a direct association with the tumour microenvironment, which has been evident in the modulation of tumour growth and activity [1]. Comparatively, secondary interactions between the tissue or organ system containing the tumour and the gut microbial community can typically serve as biomarkers for specific cancer types [1]. Under normal biological conditions as a tertiary (indirect) interaction, symbiotic gut bacterial species coordinate the action of immunomodulatory molecules that assist in the maturation of the human immune system [1]. The gut metabolites produced by these microbial communities act as signalling molecules and substrates for metabolic processes within the host, such as infiltrating the cancer cell to activate immune cells, which inhibit pro-inflammatory cytokines [12]. Specific gut metabolite types, including SCFAs, bacteriocins, phenylpropanoid-derived metabolites, and tryptophan, are of particular interest in the inhibition of carcinogenesis, as these compounds have exhibited significant activity on immune signalling and cell division processes [13]. SCFAs can recognise specific G protein-coupled receptors, including GPR41 and GPR43, on the surface of immune cells, which leads to an enhanced concentration of total regulatory T cell numbers, transforming growth factor-β (TGFβ), anti-inflammatory cytokines interleukin-10 (IL-10) within the host [14]. These findings were supplemented by observations that SCFA-rich diets demonstrated suppressive action on T-cell-mediated autoimmune responses, however, further investigation is required to identify specific metabolic targets in this association [14]. Comparatively, specific risk factors can contribute to perturbations to the immune system within the host. For instance, the regular use of antibiotics can damage the healthy gut microbial population, and impact the biological processes associated with these bacterial species. In particular, the use of antibiotics can lead to the improper development of the host immune system, as well as the depletion of healthy gut microbiota [5]. Gut microbiota plays an integral role in the development and function of the immune system and consequently can influence immune-regulated diseases [5]. Segmented filamentous bacteria (SFB) is a known immunomodulatory bacterium that has been shown to contribute to the development of autoimmune arthritis and autoimmune encephalomyelitis, which was indicative of the importance of maintaining a healthy gut microbial composition for gut homeostasis and health [5]. Existing studies have established that high concentrations of beneficial or protective bacteria in gut microbiota correlate with a more fully developed and complex immune system which can combat pathogenic microbes from the external environment better [15]. Gut microbial metabolites can enter the bloodstream of the host and impact organs distant from the gastrointestinal immune system [15]. Collectively, the close correlation between gut microbial species and the host immune system is a significant factor of consideration in the therapeutic or causative role of gut metabolites in cancer.

## 2. Methods

### 2.1. Search Strategy

The articles for the systematic literature review were sourced from PubMed, Scopus, Embase, and Web of Science. Articles were searched by using the keywords “breast neoplasms”, “breast cancer”, “breast malignancy”, “breast cancer cells”, breast cancer in vitro”, and “breast cancer in vivo”, in conjunction with “gastrointestinal microbiome”, “gut microbiome”, “gut microorganism”, “gut microbial metabolite”, “gut metabolite”, “gut microbiota”, “gut microbe”, “gut microbes”, “nisin”, “Nisin Z”, “Nisin ZP”, “bacteriocins”, “butyrate”, “sodium butyrate”, “Na Butyrate”, “butyric acid”, “inosine”, and “natural purine nucleosides”. Following the initial literature search, each article was assessed for its relevance to the current systematic literature review on the action of specific gut metabolites on breast cancer based on the following criteria: title, the information provided in the abstract, and the key focus of the overall study in relation to gut metabolites and breast cancer. The time period in the literature search was undefined and included all relevant articles published on the topic to date. An additional search strategy was used to search the reference lists of selected relevant articles to identify additional sources. These studies were also analysed with the exclusion criteria. The exclusion criteria included: book chapters, conference proceedings, food studies, retracted articles, irrelevant methodologies/outcomes in the source, restricted sources, and studies that did not discuss breast cancer or gut metabolites. The overall breakdown of the screening process is shown in the PRISMA flow diagram (Figure 3). The analysis of these resources was carried out using the summarisation of the information in the articles, inclusive of, but not limited to, the author, the purpose of the research study, and central conclusions drawn from the studies.

### 2.2. Data Extraction

The data from the included studies were extracted in a Microsoft Excel spreadsheet and the following information was captured from preclinical and clinical studies: type of gut metabolite, type of study, cancer type, cell line, the efficacy of the treatment, and molecular mechanisms of action. Similarly, the key focuses of the overall studies were extracted, as well as main conclusions drawn in relation to the association between gut metabolites and breast cancer.

## 3. Results

A total of 2724 articles were screened from the selected databases and other sources, in which 82 relevant articles were included based on the pre-determined inclusion and exclusion criteria for screening. As the literature and systematic reviews were also included in this paper, information extracted from these sources included the key focus and main conclusions of the studies, as well as general background information to support the scope of this review.

**Table 1 ijms-23-09490-t001:** A tabulated summary of studies assessing the anticancer activity of key gut metabolites in relation to breast cancer.

Metabolite Group	Metabolite	Cancer Type	Type of Study	Cancer Cell Line/Animal Type	Type of Assay	Inhibitory Effect	Reference
Bacteriocin	Nisin	Breast	In vitro	MCF7	3-(4,5-dimethylthiazol-2-yl)-2,5-diphenyl-2H-tetrazolium bromide (MTT)	High cytotoxicity with the IC_50_ value of 5 μM, and selectivity against the MCF7 cells.	[16]
	Nisin	Breast	In vitro	MCF7	MTT	Decreased cell viability in a concentration-dependent manner with the IC_50_ value of 105.46 μM.	[17]
Short-chain fatty acids	Sodium Butyrate	Breast	In vitro	MCF7	MTT	Inhibited cell proliferation in a dose-dependent manner with the IC_50_ value of 1.26 mM. Induced morphological changes to the MCF7 cells, and cell cycle arrest in the G_1_ phase.	[18]
	Sodium Butyrate	Breast	In vitro	MCF7	Cell counting kit-8 (CCK-8) and Western blot	Inhibited MCF7 cell viability in a dose- and time-dependent manner, decreased B-cell lymphoma 2 (Bcl-2) protein expression, and induced morphological changes.	[19]
	Sodium Butyrate	Breast	In vitro	MCF7 and MB-MDA-468	MTT and Annexin-V-FITC	Induced cytotoxicity and apoptosis in both breast cancer cell lines, and increased expression of 15-lipoxygenase type 1 (15-Lox-1) and production of 13-Hydroxyoctadecadienoic acid (13(S)HODE).	[20]
	Sodium Butyrate	Breast	In vitro	MCF7, T47-D, and MDA-MB-231	MTT and sulforhodamine B (SRB)	Initiated epigenetic changes to acetylation of proteins; pyruvate kinase activity was increased in MDA-MB-231 cells and lactate dehydrogenase activity was increased in T47-D cells. Increased oxygen consumption in the MDA-MB-231 and T47-D cell lines.	[21]
	Sodium Butyrate	Breast	In vitro	MCF7	CCK-8	Inhibited cell proliferation in a dose- and time-dependent manner. Induced cell cycle arrest in the G_1_/G_2_ phase and a decrease in the S phase and caused chromatin relaxation.	[22]
	Butyrate	Breast	In vitro	MCF7	Western blot and polymerase chain reaction (PCR)	Cell inhibition of 34% against MCF7 cells, increased histone H3K9 acetylation, and increased expression of p21^waf1^ and Retinoic acid receptor beta (*RARβ*).	[23]
	Sodium Butyrate	Breast	In vitro	SKBR3	MTT	Combined treatment of NaB and trastuzumab demonstrated synergistic growth inhibition and elevated mRNA and protein levels of p27^Kip1^.	[24]
	Sodium Butyrate	Breast	In vitro	MRK-nu-1	Western blot and caspase assay	Induction of caspase-3, -10, and -8, and formation of DNA fragmentation, in a dose- and time-dependent manner. Triggered apoptosis via the induction of caspase-10 activity.	[25]
	Sodium Butyrate	Breast	In vitro	MCF7	MTT	Inhibited cell growth of MCF7 cells dose-dependently, induced cell cycle arrest in the G_2_/M phase, reduced p53 expression, decreased Bcl-2 mRNA and protein levels, increased apoptosis, and reduced glutathione levels.	[26]
	Sodium Butyrate	Breast	In vitro	MCF7	Western blot and flow cytometry	Induced cell cycle arrest and apoptosis via interaction with p21^waf1/cip1^ with cyclin-dependent kinase (CDK) and decreased proliferating cell nuclear antigen (*PCNA*) levels.	[27]
	Sodium Butyrate	Breast	In vitro	MCF7, T47-D, and BT-20	Western blot	Increased the expression of tumour necrosis factor receptor 1 (TNF-R1) and receptor 2 (-R2), TRAIL receptor 1 (TRAIL-R1) and receptor 2 (-R2), and Fas in MCF7 cells and acted synergistically with these receptors to inhibit cell proliferation and induced apoptosis via p21^waf1^ and its interaction with *PCNA*.	[28]
	Sodium Butyrate	Breast	In vitro	MCF7, MCF-7ras, T47-D, BT-20, and MDA-MB-231	Western blot and PCR	Inhibited cell proliferation in all cell lines. Induced cell cycle arrest in the G_2_/M phase in MDA-MB-231 cells, and in the G_1_ phase for the other four cell lines. Inhibited cell growth in a p53-independent manner and induced apoptosis via the Fas/Fas L system.	[29]
	Sodium Butyrate	Breast	In vitro	MCF7	MTT	Increased bioavailability when coupled with the hyaluronic acid drug delivery system due to the ability to bind to CD44 receptors, which are prominent on tumour surfaces.	[30]
	Sodium Butyrate	Breast	In vitro	MDA-MB-231	Flow cytometry, Western blot, and protein array analysis	Induced cell cycle arrest in the G_2_ phase via the inhibition of histone H1 kinase activities, and increased levels of p21^waf1^.	[31]
	Sodium Butyrate	Breast	In vitro	MCF7, MDA-MB-231, T47-D, and BT-20	Flow cytometry and Burton method to assess variation of DNA content	Inhibitory effect of 85-90% with a dose- and time-dependent inhibition of cell proliferation, induced cell cycle arrest in the G_2_/M phase, resulting in the induction of apoptosis in the estrogen receptor-positive cell lines MCF7 and T47-D.	[32]
	Sodium Butyrate	Breast	In vitro	MCF7	Estrogen receptor assays	Initiated significant hyperacetylation of histones in MCF7 cells and lowered estrogen receptor levels.	[33]
	Sodium Butyrate	Breast	In vitro	MCF7	CEA-Roche and Biorad protein assay	Induced morphological changes in MCF7 cells and reduced cell proliferation.	[34]
Natural purine nucleoside	Inosine	Breast	In vitro	MCF7 and MDA-MB-231	CyQuant XTT	Demonstrated primary cytoprotective activities during breast cancer hypoxia, rather than adenosine, which was previously thought to be the primary compound responsible for this bioactivity.	[35]

**Table 2 ijms-23-09490-t002:** A tabulated summary of the clinical research on key gut metabolites for their action against different malignancies including breast cancer.

Metabolite Group	Metabolite	Cancer Type	Clinical Study Details	Clinical Observations	Reference
Short-chain fatty acids	Butyric acids, propionate, and acetate	Colorectal	A case-control study with 14 colorectal cancer (CRC) patients and 14 non-CRC subjects.	A decreasing abundance of SCFA-producing bacterium, e.g., *Bifidobacterium*, in CRC patients in comparison to non-CRC participants. The levels of all three SCFAs assessed were reduced in CRC patients, and the values for butyric acid and propionate were statistically significant.	[36]
	Acetic, propionic, butyric, valeric, and plasma isovaleric acid	Solid cancer tumours	Prospective cohort biomarker study of 52 patients with solid cancer tumours that completed programmed cell death-1 inhibitors (PD-1i) therapy.	High concentrations of all SCFAs correlated with extended progression-free survival, and it was indicated that SCFA concentrations in stool samples may be associated with PD-1i efficacy.	[37]
	Butyrate and propionate	Breast	Conducted 16S rRNA gene sequencing, cell culture methods, and targeted metabolomics on faecal samples from premenopausal breast cancer patients and premenopausal healthy participants.	The abundance of SCFA-producing bacteria and enzymes was significantly reduced in premenopausal breast cancer patients in comparison to premenopausal healthy participants, and the overall composition of the gut microbiota differed substantially between the two groups.	[38]
Bacteriocin	Azurin-p28 peptide	P53(+) metastatic solid tumours	NSC745104: Phase I human clinical trial of azurin-p28 in 15 patients (aged 47–80 years old) with p53(+) metastatic solid tumours	Participants did not exhibit an immune response to p28, significant adverse events, or dose-limiting toxicities. Indicative of a highly favourable therapeutic index for anticancer activity.	[39]
	Azurin-p28 peptide	Central nervous system (CNS) tumours	NSC745104: Phase I human clinical trial on 18 children aged 3–21 years old with progressive or recurrent CNS tumours	The p28 peptide was well-tolerated in children with CNS tumours at the recommended adult phase II dose (4.16 mg/kg/dose), which correlated closely with the previous study on adult participants. The primary adverse event was grade 1 infusion-related reactions; however, these often did not require treatment and were short-lived.	[40]

## 4. The Correlation between Gut Metabolites and Breast Cancer Development

As an emerging and niche area of research, gut microbial metabolites have demonstrated promising anticancer potential in preclinical studies. Despite the number of risk factors associated with the onset of breast cancer, multiple lifestyle factors including increased physical activity, lactation, and successful pregnancies also exhibited protective activity against breast cancer development [2]. Gut microbial species can directly modulate breast cancer risk via alterations to host metabolism, estrogen hormone recycling, and immune pressure [41]. Additionally, it has been reported that gut microbial species can translocate to the breast tissue via the skin, which may play a significant role in the maintenance of breast health [5]. This translocation is believed to occur via multiple pathways, including sexual contact, nipple-oral contact via lactation, or nipple-areolar orifices [5]. The possibility of translocation of bacterial species from the gut to breasts through systematic circulation has also been proposed [5]. Gut microbial dysbiosis can cause the breakdown of mucosal barriers enabling the gut microbial species to translocate into the peripheral circulation and the mesenteric lymph nodes (MLN) which leads to altered immune responses [5]. Our recent review underlined the therapeutic role of gut microbial metabolites SCFAs, bacteriocins, phenylpropanoid-derived metabolites, prenylflavonoids, and natural purine nucleosides in cancer [12]. The current review focuses specifically on the therapeutic and causative role of gut microbial metabolites on breast cancer.

Specific epigenetic factors have been observed to contribute to the progression or inhibition of breast cancer development. DNA methylation is an epigenetic alteration that serves as a biomarker for the early detection of breast cancer, as well as histone modifications and microRNAs, which may be useful in breast cancer treatment specific to each cancer patient [42]. These three epigenetic events have been observed to be mutually interacting in the regulation of breast carcinogenesis. It has been established that several environmental and/or lifestyle factors directly contribute to epigenetic mechanisms that trigger breast cancer development. A retrospective case-control study examined the influence and prevalence of environment and lifestyle factors on breast cancer risk in the female population in Malta [43]. That study found that breast cancer risk was reduced in individuals exposed to greater levels of sunlight and in individuals who were not taking the oral contraceptive pill [43]. However, an increased risk of breast cancer was observed with increased height in participants [43]. Further longitudinal studies are recommended to assess the various epigenetic factors associated with the development and risk of breast cancer.

### 4.1. The Microbiota of Healthy Breast Versus Breast Tumour Microenvironment

The breast tissue microbiome is constituted by a multitude of lifestyle and biological factors, including the translocation of bacterial species from the gut to the breast, sexual activity, and breastfeeding, therefore, the microbiota can be altered by changes to these factors [44]. One study implemented next-generation sequencing to investigate the potential difference in microbial composition between breast tumour tissue and paired normal adjacent tissue from one patient [3]. This study identified high concentrations *Methylobacterium radiotolerans* in the breast tumour tissue, in comparison to the paired normal tissue, which presented with high concentrations of *Sphingomonas yanoikuyae* [3]. The reduction in the *Sphingomonas* species in the tumour tissue implied a potential probiotic role within the host, and this bacterium was also found to activate invariant NKT (iNKT) cells [3]. The iNKT cells have been observed to play a significant role in the modulation of breast carcinogenesis as a cancer immunosurveillance agent, which could implicate the importance of *Sphingomonas yanoikuyae* in the regulation of breast cancer development [3]. A later study assessed potential pathogenic biomarkers in breast cancer patients, and observed that the *Methylbacterium* bacterial species was in higher concentrations in more advanced breast cancer cases, however, did not differ amongst tumour grades [45]. As the relative concentrations of each bacterial species inversely correlated with the tissue type, these studies were among the first few to acknowledge the link between microbial dysbiosis and the onset of carcinogenesis, indicating its clinical relevance in the diagnosis and staging of breast cancer [3,45]. Figure 4 depicts the gut microbiota and mammary microbiota present in healthy individuals, compared to individuals with cancer.

### 4.2. Microbial Dysbiosis and Breast Cancer Growth

Several known risk factors contribute to breast carcinogenesis, including genetic predisposition, age, and dietary intake, however, the majority of breast cancer types have an unknown etiology [3]. Two primary risk factors of breast cancer development are mutations in the *BRCA1* and *BRCA2* genes, which can be linked to a family history of breast cancer, as well as the use of hormone-replacement therapy that causes extended exposure to female hormones, such as estrogen [2]. Typically, breast cancer cells present with pathological alterations to metabolic processes, which impacts the metabolism of the host leading to increased breast cancer risk [2]. The Warburg effect explains that breast cancer cells undergo aerobic glycolysis that initiates decreased mitochondrial oxidation and increased glycolytic flux, which support the prominent proliferative action of breast cancer cells (Figure 5). However, this phenomenon has since been built upon and other metabolic pathways have been observed to be increased in breast cancer patients, including cholesterol and glutamine metabolism, lipids and fatty acids, protein translation, and the glutamine-serine pathway [2].

A recent study stated that a notable 20% of cancer cases have a causative association with gut microbial dysbiosis and pathogenic bacterial species, such as *Helicobacter pylori* [13]. The pathogenesis of different cancer types is associated with inflammation, which is triggered by the dysbiotic processes and increases exposure to pathogenic microbes within the host [46]. However, microbial dysbiosis also predisposes the host towards cancer by initiating DNA damage, genetic instability, altered response to anticancer therapies, and metabolic dysregulation [10]. A profiling study sought to conduct gut microbial profiling to determine potential microbial signatures for breast cancer, in which it was observed that there were substantial differences in gut microbial composition between breast cancer patients and healthy individuals, as well as between breast cancer survivors and healthy individuals [10]. Additionally, it was noted that the resistant bacterial and fungal species of the gut microbial communities can produce metabolites that interact with signalling pathways to mediate and influence breast cancer progression, and may directly affect drug metabolism and efficacy (pharmacokinetics, and anti-tumour toxicity) [10].

Whilst previous studies have primarily recognized the significance of the perturbations to the breast microbiota in breast cancer tumourigenesis, the consequences of alterations to the gut microbiota have been less prioritised within the literature. It has been shown that the modulation of the intestinal microbial composition can also alters gut metabolite concentrations which can both contribute to the onset of cancer and induce carcinogenesis distal from the origin site [13]. This causative effect could have several clinical implications for the development of malignancies in multiple organs or body systems. A few studies performed microbial profiling analyses to compare the gut microbiota between breast cancer patients and healthy individuals and observed that the diversity of bacterial species was substantially reduced in breast cancer patients in comparison to that of the healthy controls (Figure 4) [47,48,49,50]. In particular, the abundance of β-glucuronidase-producing bacterial species was increased in breast cancer patients, including both the *Clostridium coccoides* and *Clostridium leptum* subspecies [47,48,49,50]. β-glucuronidase-producing subspecies exhibit enzymatic activity that alter the systemic and local concentrations of the estrogen hormone, which is significant as estrogen hormone levels are elevated in breast cancer patients [51]. Similarly, the β-glucuronidase-producing bacterial species were detected in the nipple aspirate fluid of breast cancer survivors and have been found to deconjugate compounds within the host, which increases the duration they remain in circulation [51]. These observations were further supported by a subsequent review stating that the pro-tumoural effects of gut dysbiosis are primarily caused by virulence factors and specific toxins, i.e., the upregulation of toxic gut metabolites and the inhibition of protective metabolites in different cancer types [13]. The authors also identified that the presence of specific secondary bile acids (BAs) can induce carcinogenesis via multiple mechanisms, including the induction of DNA damage, activation of the β-catenin signalling pathway, and an increase in cyclooxygenase-2 (COX-2) activity [13].

Interestingly, secondary BAs have been also shown to exert anticancer activity and reduce cancer risk in preclinical studies, which is important when assessing the effects of alterations to healthy and toxic secondary BA ratios within the host [12]. One study utilised metagenomic analysis to profile the gut microbiome of pre- and postmenopausal breast cancer patients, and healthy controls, and found that both the composition and function(s) of the gut microbial species differed significantly between postmenopausal cancer patients and healthy individuals [44]. This study highlights that the gut microbiome, and consequently gut microbial dysbiosis, could serve as a biomarker for breast cancer diagnosis, prognosis, and treatment and that in several circumstances, healthy modification of the microbiota can be achieved by implementing dietary alterations [44]. Overall, the prevention or modification of microbial dysbiosis within the host is significant in the management of overall breast cancer risk.

## 5. Anticancer Action of Nisin against Breast Cancer

As the most produced bacteriocin in the gut, nisin has demonstrated strong anticancer potential in different studies, however, has not been evaluated widely against breast cancer (Table 1). Nisin is a polycyclic peptide comprised of 34 amino acids in its molecular structure and produced by Gram-positive *L. lactis* through fermentation [16,52,53]. This gut microbial metabolite is the most common bacteriocin (also known as lantibiotic) and the only bacteriocin approved by the U.S. Food and Drug Administration (FDA) for use in food applications, as it is safe for human consumption and non-toxic to animals [16,54]. Nisin demonstrated a broad spectrum of pharmacological effects, which is inclusive of the inhibition of Gram-negative bacterial species and has been used for several years in the prevention of pathogenic bacterial growth in foods [16,54]. In addition, nisin participates in a phenomenon known as ‘colonisation resistance’, in which it competes with and eliminates other Gram-positive bacterial species within the region in the gut [55]. Under normal physiological conditions, nisin triggers changes in cell membrane potential by modulating the integrity of the membrane and forming short-lived pores along the membrane [53,54]. The cationic portions of the nisin amino acids extend through the cell membrane to one side of the molecule and interact directly with the negatively charged phospholipid heads of the membrane, whilst the hydrophobic portions of nisin interact with the core of the membrane [54]. This modulation of cell membrane activity enables the influx of ions into the cell, such as calcium [54]. Additionally, this compound interacts directly with the innate immune system by increasing the secretion of chemokines and inhibiting lipopolysaccharide-stimulated cytokines as supported by both in vitro and in vivo preclinical studies [16]. The complex pharmacological properties of nisin and its role in host health have made it a promising lead for anticancer research.

To date, minimal preclinical studies have been conducted investigating the activities of nisin against different breast cancer types. However, a recent in vitro study demonstrated promising selective anticancer activity of nisin against the MCF7 breast adenocarcinoma cell line and its potential synergistic action with doxorubicin [16]. The authors identified that nisin exhibited high and selective cytotoxicity against the MCF7 cells with an IC_50_ value of 5 μM and did not exhibit any cytotoxicity against the non-cancerous HUVEC cell line [16]. Furthermore, nisin induced apoptotic cell death by initiating cell cycle arrest and catalysed a calcium ion influx across the cell membrane [16]. Notably, the authors also observed that the combination of nisin with the standard anticancer drug doxorubicin presented with stronger cytotoxic activity at sub-inhibitory concentrations demonstrating potential synergistic activity compared to the individual administration of either doxorubicin or nisin [16]. This was further supported by another in vitro study, in which the combined administration of nisin with doxorubicin displayed an overall improved treatment outcome in patients with skin cancer [52]. The potential synergistic interactions between nisin and standard chemotherapy could benefit the clinical outcome of cancer. Another in vitro study compared the cytotoxic activity between nisin and a second bacteriocin (bovicin HC5) against the MCF7 cells and observed that nisin was significantly more effective in inhibiting the cancer cells compared to bovicin HC5 [17]. Additionally, the authors reported that nisin presented with strong haemolytic activity against eukaryotic cells and increased the permeability of the phospholipid bilayer, which could be an important factor of consideration in the pharmacological effects of nisin as an anticancer agent [17]. The mechanisms of action observed by Akbari & Avand [16] were initially validated by an in vivo study assessing the activity of nisin against head and neck squamous cell carcinoma (HNSCC) cell lines, in which this gut metabolite inhibited tumourigenesis via multiple mechanisms of action, including the activation of cation transport regulator homolog 1 (CHAC1), the induction of cell cycle arrest and the initiation of apoptosis, and increased calcium efflux [54].

A recent study observed synergistic action in the co-administration of nisin and the standard drug 5-fluorouracil (5-FU) against skin cancer cells in vivo, in which the inhibition of cell proliferation and the induction of apoptosis, as well as a reduction in the abundance and size of cancer cells, were greater in the combined therapy in comparison to the mono treatments [56] (Figure 6). These findings were further validated in a more recent study that found the synergistic actions of nisin and 5-FU were enhanced in vivo when bound to a single composite nanoparticle, demonstrating a significant reduction in both tumour progression and volume [57] (Figure 6). These observations, in conjunction with the fact that bacteriocins assist in a reduction in antibiotic use within the food and pharmaceutical industries, highlight the potential of nisin in decreasing and inhibiting carcinogenic pathogens within the gut microbiota [56]. Future investigation into nisin must prioritise in vitro (multiple breast cancer cell lines) and animal studies including those with limited treatment options (e.g., triple-negative breast cancer) and understand its potential synergy with standard anticancer therapies, to provide more evidence for clinical studies. Additionally, establishing the full dose-response relationship of this metabolite is a vital consideration for application in organism models, including in vivo and clinical research.

## 6. The Duality of Sodium Butyrate in Breast Cancer

SCFAs are the most common types of gut metabolites and are primarily produced by the bacterial species *Eubacterium rectale*, *Clostridium leptum*, and *Faecalibacterium prausitzii*, as well as the lactate-utilising species *Eubacterium hallii* and *Anaerostipes* [12]. As one of the most abundant SCFAs, butyrate has presented with potential anticancer activity through different mechanisms of action against breast cancer. Sodium butyrate (NaB), the sodium salt of butyrate, has also been explored for its anticancer activity in different studies, however, the mechanisms of its action have not been well-defined. Butyrate is produced naturally as a by-product of bacterial fermentation of fermentable and non-digestible carbohydrates, such as dietary fibres, in the GIT [32,58,59]. The primary bacterial group responsible for the production of butyrate is *Firmicutes*, and the total intestinal concentration of this metabolite within the GIT often exceeds 100 mM [58]. Gut epithelial cells typically utilise butyrate as an energy source, which means its systemic circulation concentration is relatively low [14]. In preclinical research, butyrate has presented with both anti- and pro-carcinogenic activities mainly governed by its concentration, known as the ‘butyrate paradox’ [12,59]. In particular, low concentrations of butyrate induce carcinogenesis within the host, whereas higher concentrations inhibit tumourigenesis and tumour progression [12]. The diverse biological activities exhibited by butyrate warrant further investigation into a safe therapeutic dosage for use in different cancer types.

### The Anticancer Action of Butyrate against BC

In normal biological processes, butyrate plays a role in the activation of epigenetic processes, including epigenetically silenced genes in cancerous cells, such as *BAK* and *p21* [60]. It also helps in the absorption and excretion of minerals, such as zinc and iodine, which are cofactors of enzymes directly involved in the epigenetic processes [60]. In addition to the significant role of butyrate in the maintenance of host health, it exhibited substantial anticancer potential across multiple cancer types in preclinical studies. As a histone deacetylase inhibitor (HDACi), butyrate and NaB demonstrated strong anti-tumoural action against several breast cancer cell lines, including MCF7, and two TNBC cell lines, MDA-MB-231 and MDA-MB-453, in the past couple of decades [22,31,34,60,61,62,63] (Table 1). A more recent in vitro study assessed the effects of NaB on cell proliferation and the ultrastructure in the MCF7 breast adenocarcinoma cells [19]. The authors confirmed that the administration of NaB induced morphological changes to the ultrastructure of the MCF7 cells, inhibited cell proliferation and induced apoptotic cell death, however, the in-depth mechanisms of action were not determined [19]. The mechanisms associated with NaB-induced apoptosis were explored in a different study that assessed anticancer activity against the MRK-nu-1 breast cancer cell line and a significant time-dependent increase in caspase-10 mRNA expression following treatment with NaB in comparison to levels of caspase-3 or caspase-8 was observed [25]. These findings implied that NaB can induce apoptotic cell death in human breast cancer cells through caspase-10 expression as the mechanism by which [25]. An earlier in vitro study also analysed the mechanisms by which NaB was able to induce apoptosis in the MCF7 cells, and reported that NaB induced cell cycle arrest in the G_2_/M phase, inhibited the expression of p53, increased levels of p21^waf1/cip1^, upregulated mRNA Bax levels, and downregulated Bcl-2 mRNA and protein levels [26]. These findings, along with the observation that NaB-induced apoptosis correlated with depleted intracellular glutathione levels, proposed that the pro-apoptotic effects of NaB were associated with oxidative stress and glutathione depletion [26]. Another study found that the two G-protein-coupled cell-surface receptors, GPR41 and GPR43, played a vital role in the management of SCFA signalling in breast epithelial cells, as well as their stress management [64]. As a supplementary finding, this study also identified that butyrate induced an influx of intracellular calcium into the MCF7 cells [64]. Furthermore, butyrate has been implicated in modulating oncogenic signalling pathways via methylation and microRNA biological processes, as well as regulation of both extrinsic and mitochondrial apoptotic pathways [61]. A summary of the observed in vitro mechanisms of action of NaB against breast cancer cells is presented in Figure 7. Whilst NaB exhibited strong anticancer potential in previous studies, the low bioavailability of the compound prevents its implementation in clinical trials. Recent studies have emphasised that this issue could be overcome by the implementation of a nano-delivery drug carrier, as well as co-administration with other anticancer agents [61]. In addition to this, a potential mechanism to alleviate the issue with the bioavailability of NaB is the systemic injection of NaB-producing bacterial species, however, depending upon the tumour type and bacterial strain, the administration route may differ substantially [65]. Possible administration routes include systemic intravenous injection, subcutaneous, oral, and intratumoural administration, of which the former has been observed as a dominant route to ensure the effectiveness and limit the toxicity of the NaB metabolite [65].

Despite the notable anticancer action of butyrate as a potential stand-alone treatment, it has also been observed to demonstrate a synergistic activity with standard anticancer drugs in recent years (Figure 6). One study assessed the co-administration of NaB with docetaxel against lung cancer cells and identified that the combination therapy synergistically inhibited cell proliferation and promoted apoptotic cell death [66]. In specific relation to breast cancer, an earlier study assessed the anti-tumour-enhancing effects of trastuzumab (Herceptin), a recombinant monoclonal antibody, when combined with NaB against a HER2-overexpressing breast cancer cell line [24]. The in vitro study determined that the anticancer effect of NaB was significantly enhanced by the co-administration of trastuzumab, as well as elevated mRNA and protein levels of p27^Kip1^ [24]. Similarly, the combinations of retinoids and HDAC inhibitors have been implored as a potential epigenetic strategy in cancer treatment. Another in vitro study investigated the potential synergistic action of butyrate and vitamin A against the MCF7 breast cancer cells and found that vitamin A potentiated the proliferative action of butyrate at a cell proliferation inhibition of 46%, in comparison to the 34% and 10% inhibition observed with butyrate and vitamin A, respectively, as stand-alone treatments [23]. An earlier study acknowledged the potential issues with the bioavailability of NaB and proposed that hyaluronic acid could serve as a drug carrier for the gut metabolite, and is preferable due to its ability to bind to CD44 on the tumour cell surface, and could improve the anti-proliferative activity of NaB against breast cancer cells, which was further assessed in a recent review [32,67]. Another in vitro study observed significant a synergistic induction of apoptosis in the co-treatment of butyrate with tumour necrosis factor-alpha (TNF- α), anti-Fas agonist antibody, or TNF-related apoptosis-inducing ligand (TRAIL) against the MCF7, T47-D, and BT-20 breast cancer cell lines [28]. The co-treatment of butyrate with these death receptors upregulated proliferating cell nuclear antigen (PCNA) and levels of P21^waf1^, indicating their beneficial interactions to potentiate apoptotic cell death as observed in the co-treatments, which could have a number of clinical implications in managing breast cancer [28]. Furthermore, another in vitro study analysed the effect of butyrate in combination with the standard anticancer drug doxorubicin against melanoma cells by implementing a conjugate targeted delivery system [68]. The authors reported that butyrate improved the overall efficacy of doxorubicin, limited the potential of drug resistance, and was more specific in targeting the melanoma tumour microenvironment [68].

A recent preclinical study also aimed at assessing the combined administration of butyrate with 5-FU, against colon cancer cells, and found that butyrate significantly increased the efficacy of 5-FU against the cancer cells, as well as mitigated the DNA synthesis impairment caused by the standard chemotherapeutic drug (Figure 6) [69]. Based on current preclinical findings, butyrate may improve the clinical efficacy and impede the cytotoxicity of standard chemotherapeutic drugs due to its action as an HDAC inhibitor. A metformin derivative, metformin-butyrate, also exhibited anti-tumour action via cell cycle arrest in both the G_2_/M and S phases and selective cytotoxicity against resistant and aggressive breast cancer cells [62]. This derivative further showed synergistic activity with the standard anticancer drugs docetaxel and cisplatin by significantly decreasing xenograft breast tumour volumes [62]. These findings were supported by a more recent in vitro study that examined the combined effects of NaB and docetaxel against A549 lung cancer cells and confirmed that the cumulative effects of the combination therapy were greater than the additive effects of each stand-alone treatment [66]. Based on existing findings, butyrate and NaB may provide substantial benefits as combination therapies with standard anticancer drugs.

Although the preclinical evidence supports the use of NaB as an anticancer agent, it also demonstrated carcinogenic activity in some studies. An in vitro study aimed to investigate the dual effects of NaB on hepatocellular carcinoma (HCC) cells and observed that the HCC cells demonstrated higher cell growth when treated with low levels of NaB (<0.5 mM) in comparison to cancerous cells that were not treated with the compound (*p* < 0.05) [59]. This study found that when treated with higher concentrations of NaB (>0.5 mM), the HCC cells underwent cell cycle arrest in the G_1_ phase and inhibition of cell proliferation. It has been shown that these low concentrations of butyrate initiate a pro-inflammatory microenvironment that perturbs the gut microbial population by encouraging the colonisation of butyrate-producing bacterial species [14]. Butyrate was also shown to downregulate the lipopolysaccharide-mediated proinflammatory factors such as nitric oxide, IL-6, and IL-12 produced by macrophages via the inhibition of histone deacetylases [70]. This study indicated that butyrate in the gut renders lamina propria macrophages less responsive to the gut microbial species through the downregulation of proinflammatory mediators [70]. Additionally, whilst not directly related to breast cancer, it has been observed that long-term exposure to butyrate by colon cancer cells results in the development of resistance to butyrate by the tumour cells, which may lead to chemoresistance in colon cancer [71]. As low concentrations of butyrate demonstrate pro-tumoural action against cancer cell lines, determining a safe therapeutic dose for administration in clinical studies is a vital factor of consideration.

## 7. Potential Implementation of Inosine in Breast Cancer Therapy

Inosine, a natural purine nucleoside, has not been assessed in preclinical research as an anticancer agent against breast cancer. Adenosine (ADO) is the final product of adenosine triphosphate (ATP) hydrolysis and has been observed to increase in concentration within the tumour microenvironment [72]. Inosine is formed via the metabolic conversion of ADO by the adenosine deaminase (ADA) enzyme and can be produced both intra- and extracellularly within the host [72,73]. This factor enables inosine to influence receptor-independent pathways and alter cell function, as well as initiate signalling events within the cell, via binding to the adenosine receptor (AR) [72].

*Bifidobacterium pseudolongum* has been shown to undergo metabolic processes to form the metabolite inosine (Figure 2), which is found in the highest concentrations in the duodenum of the small intestine, and this level decreases along the GIT [74]. This was further supported by the in vivo finding, indicating that the synthesis of inosine in the upper GIT was anticipated to be the primary source of systemic inosine concentrations in mice monocolonised with *B. pseudolongum* [74]. To further support these data, dose-response experiments determined that the higher concentrations of inosine observed in *B. pseudolongum* sera were sufficient to initiate T_H_1 activation in vitro, and inosine increased the levels of immune factors in the MLN in vivo [74]. In the absence of IFN-γ, inosine inhibited T_H_1 cell differentiation, whereas in the presence of IFN-γ, inosine was able to accelerate the differentiation of T_H_1 from undifferentiated T cells [74]. The influence of inosine on T cell differentiation was dependent upon co-stimulation by other immune factors, sufficient IFN-γ production to increase anticancer immunity, and IL-12 receptor interaction to ensure T_H_1 differentiation by inosine [74].

The potential use of inosine as a natural anticancer agent is based on the observed activity of this metabolite on the efficacy of immune checkpoint blockade (ICB) therapy, and the administration of certain inosine-producing bacterial species can increase immunity and the efficacy of ICB therapies [74]. An in vitro study assessed the activity of inosine on human C32 melanoma cancer cells and observed that in micromolar concentrations inosine induced cell proliferation via A_3_AR activation [72]. The A_3_AR antagonist is upregulated in different tumour types and is identified as exhibiting tumour-specific and distinct anticancer activity, which is dependent upon the tumour type [72]. Activation of the A_3_AR antagonist by inosine caused an increase in ERK1/2 levels, as well as the activation of P2Y_1_R via ENT-dependent mechanisms of action, and it was proposed that the proliferative activity induced by inosine was caused by the simultaneous activation of PI3K and PLC-PKC-MEK1/2-ERK1/2 pathways [72]. Whilst these findings are significant in understanding the role of inosine in melanoma cancer progression, the mechanisms associated with this activity are not understood or addressed in the existing literature across a variety of different cancer types. A more recent study evaluated the role of inosine in the occurrence of breast cancer hypoxia, and the data indicated that inosine is the primary cytoprotective compound during this process, which contradicts the previous concept that adenosine was the primary compound [35]. The bioactivity conferred by inosine during breast cancer hypoxia warrants further investigation to determine the exact mechanisms of action. Additionally, drug resistance to standard chemotherapy is an ongoing issue in oncological research and may require a combined therapeutic regime with natural agents to combat drug resistance [75]. Given the limited evidence available, inosine requires further investigation to establish a more holistic understanding of its anticancer potential against breast cancer.

## 8. Gut Microbial Metabolites and Clinical Research in Breast Cancer

### 8.1. Clinical Studies Exploring the Association between Gut Metabolites and Cancer Development

Despite the preclinical evidence supporting the association between gut metabolites and breast cancer, clinical studies are limited (Table 2). A currently ongoing case-control clinical study (NCT03885648) will provide the first clinical evaluation into the association between gut microbial species, dysbiosis, and the associated risk of breast cancer, which could develop further understanding into determining novel interventions for breast cancer and improving its overall prognosis [76]. The overall breast cancer risk was hypothesised to be correlated with both the composition and functionality of the intestinal and mammary gut microbiomes [76].

To assess the potential correlation versus causation between the presence of SCFAs and cancer progression, a clinical study profiled the microbial composition of CRC patients and observed that the concentrations of SCFAs, including butyrate, acetate, and propionate, were substantially reduced in CRC patients in comparison to the non-CRC control group [36]. This observation was in conjunction with the finding that the microbial profile of *Bifidobacterium* species differed significantly between the CRC patients and non-CRC participants, and the predominant group of *Bifidobacterium* species was absent in CRC patients [36]. The distinguishing features of the gut microbial profiles between the two groups underlined the role of gut microbial metabolites and microbial dysbiosis in the prevention or onset of cancer development. Additionally, a clinical cohort study assessed the faecal and plasma SCFA concentrations in patients with primary cancer (solid tumours) treated with programmed cell death-1 inhibitors (PD-1i; immune checkpoint inhibitors) nivolumab and pembrolizumab to examine if the gut microbiome is a contributing factor to immune checkpoint inhibitor efficacy [37]. The authors reported that faecal SCFA levels may correlate with PD-1i efficacy, which may serve as a causative link between the gut microbiome and immunotherapy, as well as a potential routine monitoring measure for cancer patients [37]. Of particular significance was a clinical study which measured alterations to intestinal microbial species of premenopausal breast cancer patients, in which the concentrations of two SCFA-producing bacterial species, *Pediococcus* and *Desulfovibrio*, were distinguishable between the healthy premenopausal women and premenopausal breast cancer patients [38]. The abundance of the two SCFA-producing bacteria was substantially reduced in the premenopausal breast cancer group, as well as the key enzymes involved in the production of SCFAs, which implied the potential of *Pediococcus* and *Desulfovibrio* as diagnostic biomarkers for premenopausal breast cancer [38]. As such, clinical studies have acknowledged the potential role of SCFAs in the detection and inhibition of cancer development, which is an important consideration for future research studies.

Nisin has been minimally investigated in preclinical research and has not yet been implemented in clinical studies. One review sought to summarise the existing preclinical and clinical studies being conducted on the anticancer action of bacteriocins [77]. To date, the only bacteriocin to enter phase I of a clinical trial is azurin-p28 peptide, which is a drug used in chemotherapy treatment and it demonstrated substantial activity in preclinical pharmacological studies [39,40]. Preclinical research supported the evaluation of this compound in a clinical setting, and azurin-p28 was observed to show selective cytotoxicity to cancer cells and inhibited cancer cell growth via the induction of apoptosis and inhibition of cell proliferation [78,79]. The primary goal of a two-part registered phase I clinical study (NSC745104) was to observe and determine the optimal safe therapeutic dose and potential adverse effects of azurin-p28 in treating adult patients with solid tumours and pediatric patients with advanced or recurring CNS malignancies by inhibiting p53 ubiquitination [39,40]. Azurin-p28 did not cause severe adverse reactions or present with cytotoxicity even at the highest tested concentration of 4.16 mg kg^−1^ and was well-tolerated in both adult and pediatric patients with solid tumours and recurrent CNS malignancies, respectively [39,40] (Table 2). However, whilst multiple bacteriocins have been preclinically evaluated and azurin-p28 is in the process of being clinically evaluated, there are a number of limitations with their administration, including bioavailability, solubility within an in vivo model, susceptibility to proteolytic enzymes, and biological stability, as well as restrictions to large-scale production for clinical application [80]. However, advances in bioengineering technologies have enabled the development of semi-synthetic and synthetic compounds that could achieve optimised stability and pharmacokinetic profile, in addition to improved bioavailability within the host and increased distribution and elimination rates [80]. Based on existing clinical research into azurin-p28, further preclinical research is required to support the implementation of nisin and other bacteriocins into clinical trials.

Natural purine nucleosides have not been explored in clinical studies, as there is limited preclinical evidence on their potential anticancer activity. Specifically, inosine has been tested in vitro and in vivo in only two preclinical studies, which, despite obtaining promising data, did not quantify a safe therapeutic dosage for human administration [72,74]. Nonetheless, the investigation by Mager et al. [74] observed that inosine modulated the host response to checkpoint inhibitor immunotherapy. Extensive preclinical research is necessary to support the implementation of the inosine into phase I clinical trials as an anticancer agent.

### 8.2. Gut Metabolites and Standard Chemotherapies

Studies have indicated that gut microbes can influence the outcome of standard chemotherapy. In recent years, studies have observed that healthy gut microbial populations have a direct impact on the efficacy and toxicity of chemotherapy and are also disturbed by chemotherapeutic administration [81]. As the composition of the gut microbiota impacts the outcome of standard chemotherapeutics, disruption to this population by chemotherapy-induced microbial dysbiosis can have a number of clinical consequences for treatment efficacy [75]. Recent reviews have emphasised the potential of targeted modulation of the gut microbiome as an additional measure in the treatment regime of oncology patients, in which the manipulation of the gut microbial composition may increase the therapeutic potential of existing anticancer drugs [82]. This may be achieved by enhancement or depletion of specific microbial species, as well as the addition of absent communities that may benefit the cancer treatment [82]. To support this proposition, a more recent review acknowledged the restoration of gut microbiota homeostasis and a decrease in systemic estrogen levels as a potential therapeutic approach to reduce the overall risk and progression of breast cancer [83]. Further investigation should prioritise the potential benefits associated with the combined administration of gut microbial metabolites and standard anticancer drugs, which could pave new avenues in cancer treatment.

## 9. Conclusions and Future Directions

Research on therapeutic and preventative properties of gut microbial metabolites against cancer is still in its early stages. Preclinical research has found that specific gut metabolites exhibit anticancer action against different cancer types. However, studies assessing the complex crosstalk between gut microbial metabolites, the host health and immune system, and breast cancer development and treatment have not been adequately investigated. In specific relation to breast cancer, NaB has been explored for its anticancer activity, and possible mechanisms of action have been proposed in recent studies. However, there are some disparities across these studies, especially regarding safe dosages and the molecular pathways involved against breast cancer cells. More preclinical research is necessary to establish safe therapeutic dosage, potential adverse effects, bioavailability, and the way the metabolite function under different physiological conditions. Comparatively, nisin and inosine have been less explored against human breast cancer in vitro and warrant further investigation. Despite the number of studies completed on gut microbial metabolites and their associated health benefits, significant gaps in the literature still exist regarding their potential role as anticancer agents for use as stand-alone or combination therapies. In conducting this further research, the full dose-response relationship between the metabolites and breast cancer progression can be established, which is a crucial factor for enabling the application of gut metabolites in animal and clinical studies. Further investigation into this knowledge gap could potentially revolutionise clinical cancer treatments, as gut microbial metabolites could be more cost-effective, have greater health benefits with lesser side effects, be highly selective and more bioavailable than standard chemotherapy for breast cancer.

## Figures and Tables

**Figure 1 ijms-23-09490-f001:**
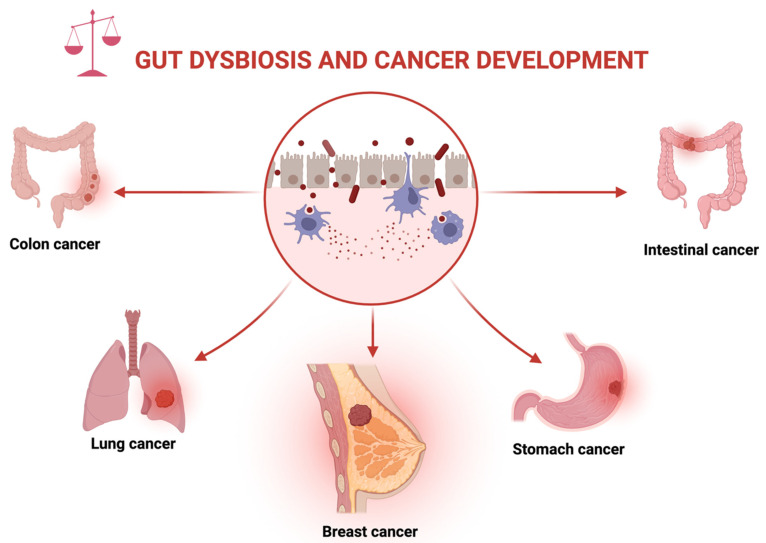
Dysbiosis (imbalance of the health gut microbial population) can lead to the development of different cancer types, including breast, colon, lung, stomach, and intestinal cancers.

**Figure 2 ijms-23-09490-f002:**
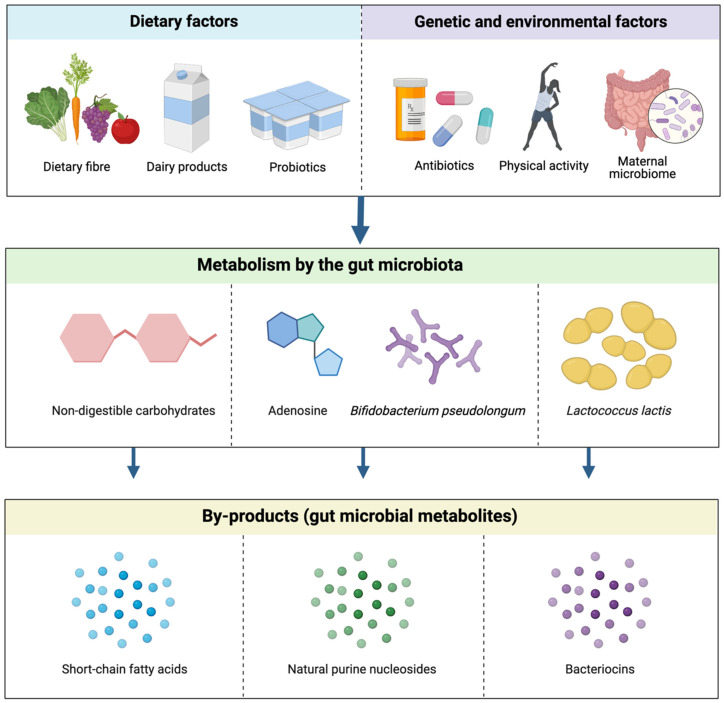
The effect of dietary and other epigenetic factors on the production of gut microbial metabolites. These factors possess precursor compounds and bacterial species, i.e., non-digestible carbohydrates, adenosine and *Bifidobacterium pseudolongum* and *Lactococcus lactis*, that undergo metabolic processes to synthesise common gut microbial metabolites, SCFAs, natural purine nucleosides, and bacteriocins, respectively [4].

**Figure 3 ijms-23-09490-f003:**
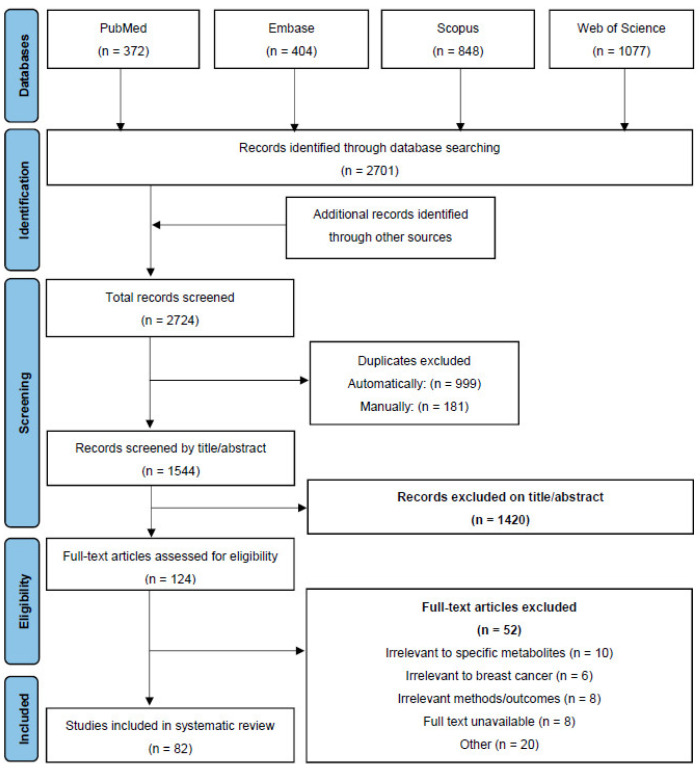
PRISMA Flow Diagram depicting the screening process for the review.

**Figure 4 ijms-23-09490-f004:**
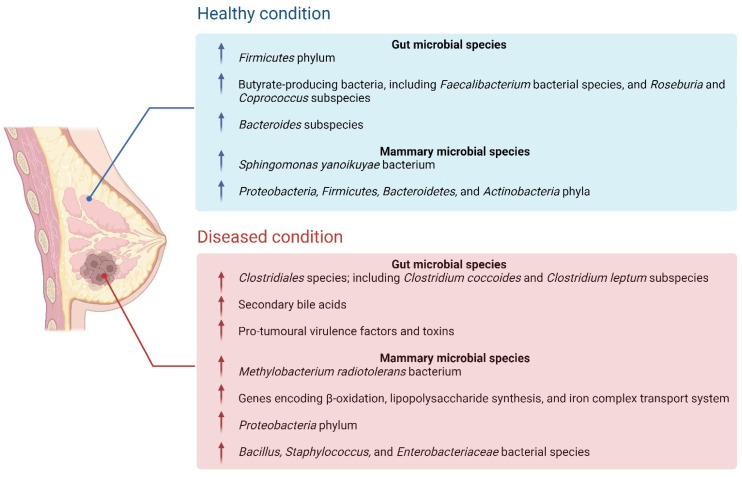
The differences in gut microbiota and mammary microbiota present in healthy individuals, in comparison to a diseased (cancerous) state. This includes increases or decreases in the abundance of protective or pro-tumoural bacterial species and subspecies between the two states.

**Figure 5 ijms-23-09490-f005:**
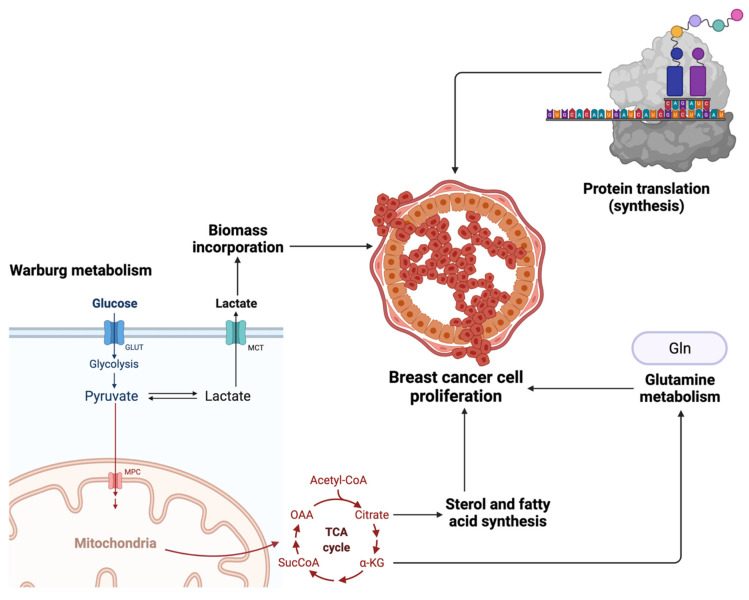
The multiple proposed metabolic processes involved in breast cancer cell proliferation, including the Warburg metabolism theory, sterol and fatty acid synthesis, glutamine metabolism, and protein translation.

**Figure 6 ijms-23-09490-f006:**
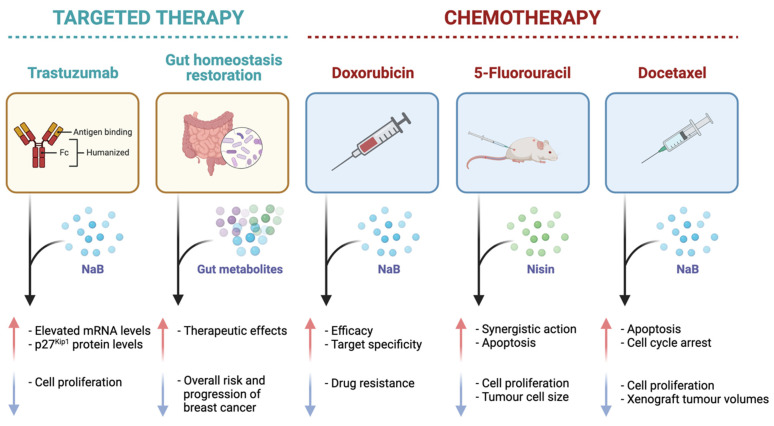
A diagrammatic representation of the effects of specific gut metabolites, sodium butyrate (NaB) and nisin, on common targeted cancer therapies, trastuzumab and gut homeostasis restoration, and standard chemotherapies (doxorubicin, 5-fluorouracil, and docetaxel) as evident in in vitro and in vivo studies in the literature. The red arrow is indicative of an increase in a certain effect, and the blue arrow is representative of a decrease in a certain effect.

**Figure 7 ijms-23-09490-f007:**
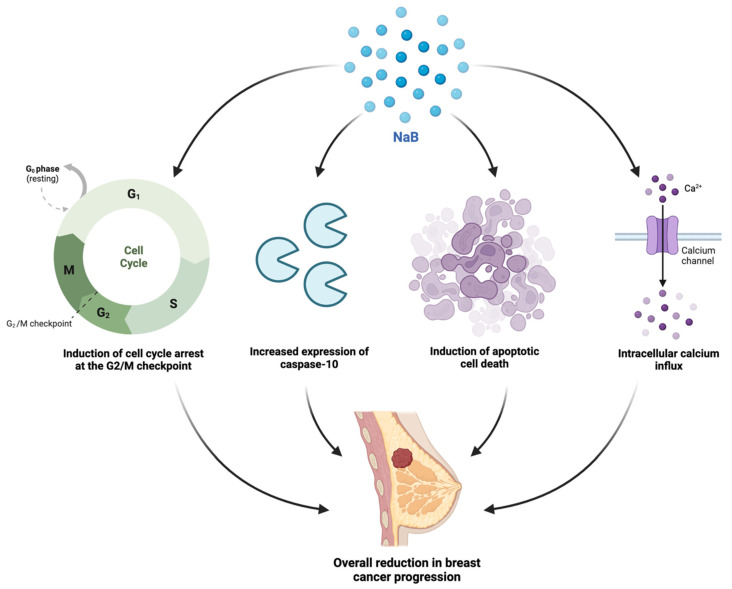
A schematic depiction of the observed in vitro molecular mechanisms of action of the sodium butyrate (NaB) against breast cancer cells including the induction of cell cycle arrest in the G_2_/M checkpoint phase, an increased expression of caspase-10, the induction of apoptosis, and the initiation of intracellular calcium influx [19,25,26,64].

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
