# Peer review of "Gut Metabolites and Breast Cancer: The Continuum of Dysbiosis, Breast Cancer Risk, and Potential Breast Cancer Therapy"

_ijms, 2022, doi:10.3390/ijms23169490_

Round 1

Reviewer 1 Report

A very nice manuscript to delineate the interaction between gut microbiota-derived metabolites and breast cancer, it is good to be published in IJMS.

Author Response

Gut metabolites and breast cancer: the continuum of gut microbial dysbiosis, breast cancer risk, and potential breast cancer therapy

We thank the Reviewers for their supportive comments and valuable suggestions. We are addressing their queries point-by-point basis below. We have made necessary changes in the manuscript that are highlighted in red.

Response to Reviewer 1

A very nice manuscript to delineate the interaction between gut microbiota-derived metabolites and breast cancer, it is good to be published in IJMS.

We thank the Reviewer for their positive feedback and recommendation.

Response to Reviewer 2

In this review, authors investigated the potential roles of gut microbial metabolites in breast cancer and its therapy. However, many summaries or conclusions are from other review papers, the data conclusion should be from the original research papers. 

We thank the Reviewer for their recommendation. We have now added four new references which are the original research papers (Ref 39, 40, 42 and 71) and relevant information to the manuscript. These newly added references are highlighted in red in the main text and reference list sections of the manuscript.

In addition, some recent studies also showed that translocated gut microbiota and their metabolites can also impact breast cancers, which can be discussed in the manuscript.

We highly appreciate this comment from the Reviewer. We have now added additional information and a new reference (Ref 42) about various ways gut microbial species can translocate to the breast tissue. These additions are highlighted in red in the manuscript (Page 14; line numbers 279 to 287)

Abbreviations in Table and context (e.g., COX-2) should be given the full names.

We have now added the full forms of different abbreviations including COX-2 throughout the manuscript (in the Tables and main text). These changes are highlighted in red.

Other changes

  • We have also added Acknowledgement, Conflicts of Interest and Author Contributions statements to the manuscript that were missing from the original submission (marked in red).
  • We have also slightly changed Figures 1, 4 and 7 to make them visually more appealing.

Reviewer 2 Report

In this review, authors investigated the potential roles of gut microbial metabolites in breast cancer and its therapy. However, many summaries or conclusions are from other review papers, the data conclusion should be from the original research papers.  In addition, some recent studies also showed that translocated gut microbiota and their metabolites can also impact breast cancers,  which can be discussed in the manuscript.

Abbreviations in Table and context (e.g., COX-2) should be given the full names.

Author Response

(The authors gave the same response as above.)
